# Unexpected arousal modulates the influence of sensory noise on confidence

**Micah Allen[1,2]\*, Darya Frank[1,3], D Samuel Schwarzkopf[1,4], Francesca Fardo[1,5,6], Joel S Winston[1,2], Tobias U Hauser[2,7], Geraint Rees[1,2]**

[1]Institute of Cognitive Neuroscience, University College London, London, United Kingdom; [2]Wellcome Trust Centre for Neuroimaging, University College London, London, United Kingdom; [3]Division of Neuroscience and Experimental Psychology, University of Manchester, Manchester, United Kingdom; [4]Experimental Psychology, University College London, London, United Kingdom; [5]Interacting Minds Centre, Aarhus University, Aarhus, Denmark; [6]Danish Pain Research Center, Aarhus University Hospital, Aarhus, Denmark; [7]Max Planck University College London Centre for Computational Psychiatry and Ageing Research, London, United Kingdom

**Abstract** Human perception is invariably accompanied by a graded feeling of confidence that guides metacognitive awareness and decision-making. It is often assumed that this arises solely from the feed-forward encoding of the strength or precision of sensory inputs. In contrast, interoceptive inference models suggest that confidence reflects a weighted integration of sensory precision and expectations about internal states, such as arousal. Here we test this hypothesis using a novel psychophysical paradigm, in which unseen disgust-cues induced unexpected, unconscious arousal just before participants discriminated motion signals of variable precision. Across measures of perceptual bias, uncertainty, and physiological arousal we found that arousing disgust cues modulated the encoding of sensory noise. Furthermore, the degree to which trial-by-trial pupil fluctuations encoded this nonlinear interaction correlated with trial level confidence. Our results suggest that unexpected arousal regulates perceptual precision, such that subjective confidence reflects the integration of both external sensory and internal, embodied states.

\*For correspondence: micah. allen@ucl.ac.uk

**Competing interests:** The authors declare that no competing interests exist.

## Introduction

Our subjective feeling of confidence enables us to monitor experiences, identify mistakes, and adjust our decisions accordingly. It is therefore crucial to understand what underlies this feeling; for example, does only the quality of available sensory signals matter, or do our confidence reports also reflect internal bodily states, such as arousal? Although confidence is thought to depend upon the quality or strength of sensory evidence, convergent computational theory and experimental data highlight the role of interoceptive inferences in guiding exteroceptive awareness. In this sense, confidence may be a metacognitive integration of both internal and external sources of uncertainty. Here, we address this possibility using a novel psychophysical design, in conjunction with signal theoretic modelling of confidence, to assess the degree to which sensory uncertainty depends upon unexpected arousal.

Computationally, confidence is typically described as the output of a feed-forward ideal statistical observer monitoring sensory (or decision) evidence. Confidence is thus determined solely by the quality or strength of sensory inputs relative to a late-stage criterion or threshold. For example, in signal detection theory, sensory samples whose average intensity fall beyond a confidence criterion are ascribed a higher certainty (*Galvin et al., 2003*; *Lau and Rosenthal, 2011*; *Maniscalco and Lau,*

**eLife digest** As you read the words on this page, you might also notice a growing feeling of confidence that you understand their meaning. Every day we make decisions based on ambiguous information and in response to factors over which we have little or no control. Yet rather than being constantly paralysed by doubt, we generally feel reasonably confident about our choices. So where does this feeling of confidence come from?

Computational models of human decision-making assume that our confidence depends on the quality of the information available to us: the less ambiguous this information, the more confident we should feel. According to this idea, the information on which we base our decisions is also the information that determines how confident we are that those decisions are correct. However, recent experiments suggest that this is not the whole story. Instead, our internal states – specifically how our heart is beating and how alert we are – may influence our confidence in our decisions without affecting the decisions themselves.

To test this possibility, Allen et al. asked volunteers to decide whether dots on a screen were moving to the left or to the right, and to indicate how confident they were in their choice. As the task became objectively more difficult, the volunteers became less confident about their decisions. However, increasing the volunteers' alertness or "arousal" levels immediately before a trial countered this effect, showing that task difficulty is not the only factor that determines confidence. Measures of arousal – specifically heart rate and pupil dilation – were also related to how confident the volunteers felt on each trial. These results suggest that unconscious processes might exert a subtle influence on our conscious, reflective decisions, independently of the accuracy of the decisions themselves.

The next step will be to develop more refined mathematical models of perception and decision-making to quantify the exact impact of arousal and other bodily sensations on confidence. The results may also be relevant to understanding clinical disorders, such as anxiety and depression, where changes in arousal might lock sufferers into an unrealistically certain or uncertain world.

*2012*). Similarly, ballistic accumulation models suggest that confidence relates to the speed of evidence accumulation relative to decision threshold (*Kiani and Shadlen, 2009*; *Kiani et al., 2014*). In both cases, confidence is generated by the bottom-up read-out of sensory information relative to a decision variable, and is assumed to depend on the same information underlying the accuracy of the perceptual decision itself.

However, emerging evidence suggests that confidence can be influenced independently of choice accuracy; for example magnetic stimulation of the motor cortex specifically disrupts confidence but not accuracy for perceptual choice (*Fleming et al., 2015*). Similarly, increased sensory noise reduces confidence even when difficulty is equated (*Spence et al., 2016*). A potential physiological mediator of these effects is bodily arousal, which regulates affective salience and perceptual variability (*Critchley et al., 2001*; *Murphy et al., 2014b*). Sudden increases in arousal trigger a reciprocal cascade of central, autonomic, and peripheral responses in the brain, heart, and pupil. Centrally, arousal is mediated by a reciprocal noradrenergic network with projections throughout the prefrontal, sensory, and limbic cortices (*Aston-Jones and Cohen, 2005*; *Murphy et al., 2014a*). This network of areas is also important for integrating perceptual and interoceptive signals (*Singer et al., 2009*; *Critchley and Harrison, 2013*; *Salomon et al., 2016*), error-awareness (*Fiehler et al., 2004*; *Klein et al., 2013*), and expected confidence or volatility (*Iglesias et al., 2013*; *Schwartenbeck et al., 2015*).

While substantial evidence supports the integration of arousal and sensory information, these observations are difficult to reconcile with ideal observer models. In contrast, predictive coding emphasizes interoceptive inference, in which confidence reflects the precision (or inverse variance) of a higher-order belief about both internal states and external sensations (*Friston and Kiebel, 2009*; *Clark, 2015*). Neurobiologically, precision is encoded by the gain of local pyramidal cells (*Bastos et al., 2012*), which is regulated across the cortical hierarchy by neuromodulators such as dopamine and noradrenaline (*Feldman and Friston, 2010*; *Friston et al., 2012*; *Moran et al., 2013*;

*Kanai et al., 2015*). The global regulation of precision by neuromodulatory gain control entails that unexpected changes in internal states should influence the estimation of confidence for other, exteroceptive channels. Predictive coding thus hypothesizes that the weight given to sensory noise depends upon expected interoceptive states, such as arousal and cardiac acceleration (*Gu et al., 2013*; *Seth, 2013*; *Barrett and Simmons, 2015*).

On this basis, we reasoned that an unexpected increase in arousal should reduce the influence of sensory noise on confidence. To test this hypothesis, we presented effectively salient, masked disgust cues in advance of a visual stimulus of variable sensory precision. Crucially, performance was equated across conditions such that changes in subjective uncertainty could be attributed to a precision-weighting mechanism, independently from any effect on choice accuracy. We further modelled evoked physiological responses (heart rate and pupil dilation), to determine whether the encoding of sensory noise in these measures also depended upon cue-induced 'arousal prediction error' (APE), and if this encoding was reflected in the trial-by-trial fluctuations of subjective confidence.

## Results

### Overview

To test these hypotheses, 29 participants performed the motion discrimination task illustrated in *Figure 1*. On every trial a global motion stimulus was preceded by a masked disgust or neutral cue. Participants then discriminated the average direction of a cloud of moving dots and rated their confidence in this decision. We used disgusted faces as arousal cues as they signal salient interoceptive and affective challenge (*Chapman and Anderson, 2012*), and elicit increased arousal and physiological responses, including heart rate acceleration and facial mimicry, even when presented without awareness (*Vrana, 1993*; *Phillips et al., 1997*; *Dimberg et al., 2000*; *Chapman and Anderson, 2012*). Furthermore, all faces were masked from awareness, allowing us to discount any role of conscious demand characteristic in our cue-related effects.

To assess the independent influence of sensory variance (or precision), the average mean and variance of motion signals were manipulated orthogonally (see *Figure 2A*) using a global-motion stimulus (see *Spence et al., 2016* for a similar technique). Crucially, to preclude an impact of task difficulty on confidence, discrimination performance was held constant (71% for low-variance trials) by adaptively adjusting the mean motion signal across trials (*Figure 2B*). Finally, to quantify the impact of sensory noise and disgust cues on perceptual choice and uncertainty, we applied a signal-

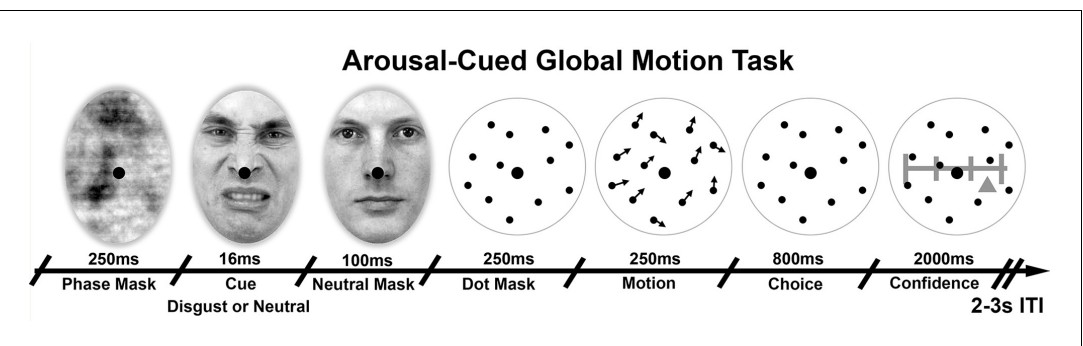

**Figure 1.** Arousal-Cued global motion task. Trial schematic illustrating our arousal-cued global motion task, in which an unexpected, masked disgusted face increased arousal just prior to a motion judgement and confidence rating. On each trial motion stimulus of variable precision (15 or 25 degrees standard deviation, σ) were preceded by either a masked disgust or neutral face, followed by the perceived neutral mask. Participants then made a forced-choice motion discrimination and subjective confidence rating. Histogram and average luminance-matching was applied between conditions and frames to eliminate pupillary artefacts, see Materials and methods for more details.
Face stimuli images taken from the Karolinska Directed Emotional Faces database and adapted with permission (ID AM25DIS) (© copyright Lundqvist D, Flykt A, Öhman A. The Karolinska directed emotional faces (KDEF) [CD-ROM]. Stockholm. Department of Clinical Neuroscience Psychology Karolinska Institutet, 1998).

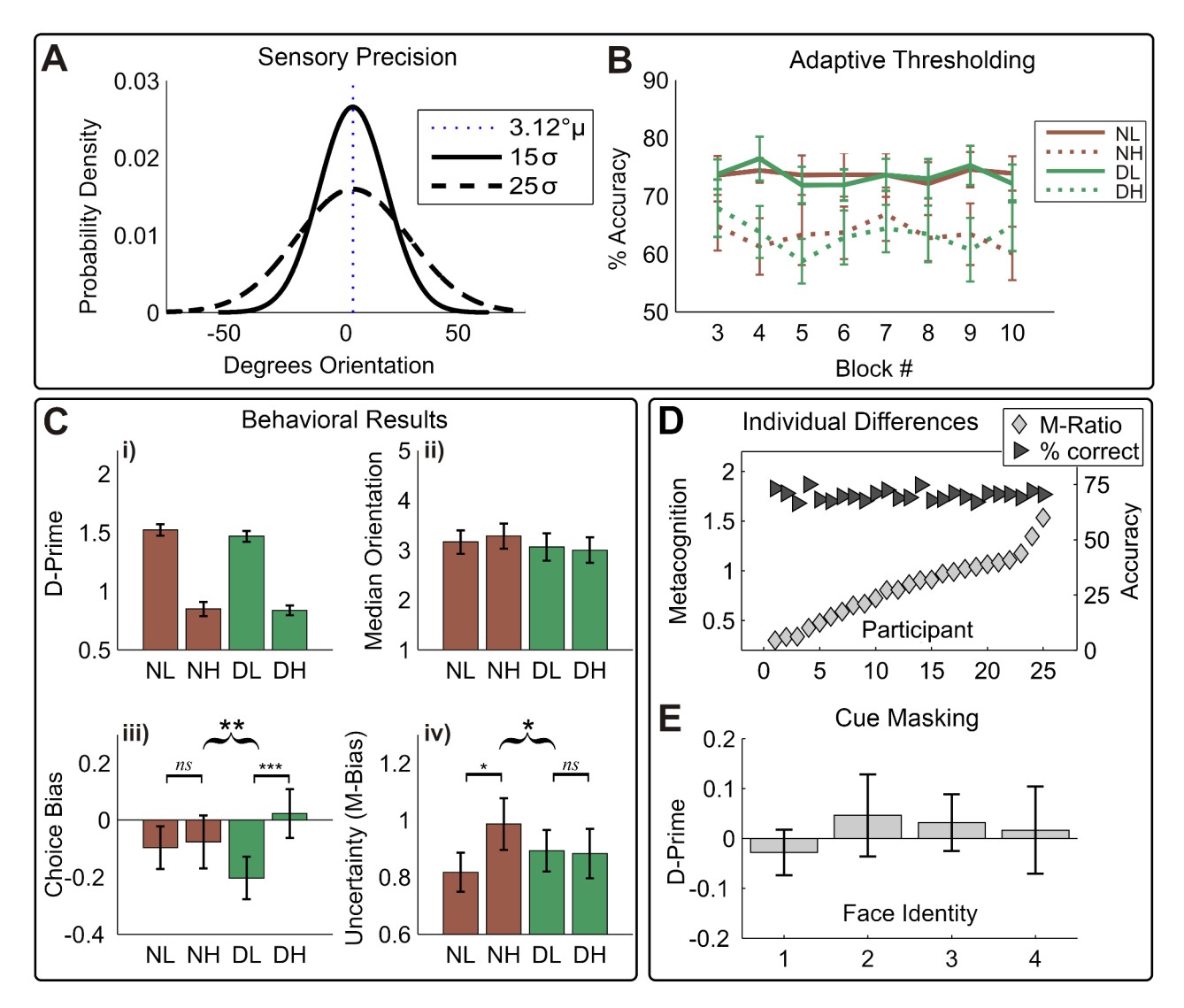

**Figure 2.** Overview of behavioral results. (A) Manipulation of sensory precision; stimulus probability density functions show our low (15 σ) and high (25 σ) variance conditions; stimulus mean and variance were orthogonally manipulated using a global-motion stimulus. (B) The performance was held constant using adaptive thresholding separately for disgust vs. neutral trials; conditions labels are neutral low variance (NL), neutral high variance (NH), disgust low variance (DL), disgust high variance (DH). (C) Degraded sensory precision reduces perceptual sensitivity; cues had no impact on either motion detection (i) or threshold (ii). Instead, disgust cues selectively increased rightward bias for low-variance stimuli (iii), suggesting arousal altered stimulus expectations. As predicted by interoceptive inference, arousing cues significantly decrease the impact of noise on uncertainty (M-bias) (iv). Curly brackets indicate F-test of 2-way interaction, square brackets indicate post-hoc t-tests (*** p<0.001, ** p<0.01, * p<0.05). All error bars +/- SEM. (D) Although performance was held constant (dark triangles, % correct), participants show considerable variability in metacognitive sensitivity (light diamonds, M-Ratio), reproducing previous results using the signal-theoretic confidence model. (E) Participants had no awareness of cue valence in a post-task forced choice test using identical trial parameters; 95% confidence intervals for d-prime on all four face pairs overlap zero (see Materials and methods, Valence Detection Task).

The following source data is available for figure 2:

**Source data 2.** Table with variable codes used in *Figure 2—source data 1*.

**Source data 1.** This csv table contains the data for *Figure 2*.

theoretic approach to modelling confidence reports (*Galvin et al., 2003*; *Maniscalco and Lau, 2012*).

## Detection performance and thresholding

In a series of control analyses, we confirmed that (1) staircases were stable across trials and between conditions, (2) staircases successfully controlled for potential cue-induced differences in detection difficulty, and (3) the masking procedure successfully prevented the detection of cue valence. Analysis of detection accuracy across blocks showed that our adaptive staircases successfully held performance stable across blocks; (F(1,24), all conditions Ps >0.12, *Figure 2B*). Further, cues exerted no influence on motion discrimination sensitivity (d-prime, d′), reaction times, or motion thresholds (*Figure 1B*, i-ii, all ps > 0.1), demonstrating that cues did not distract participants from the upcoming motion signal or otherwise alter stimulus sensitivity or detection performance. Analysis of d-prime for our forced-choice valence detection task at the end of the experiment showed that cues were not seen by participants (all 95% CIs span zero, *Figure 1D*).

Replicating previous results (*de Gardelle and Summerfield, 2011*; *Spence et al., 2016*), increased sensory noise (motion variance) rendered motion discrimination more difficult, slowing reaction times (median RT, main effect Variance, F(1,24) = 4.76, p=0.039, partial $\eta^2$ = 0.17,) and decreasing sensitivity (d′, main effect Variance, F(1, 24) = 185.15, p<0.001, partial $\eta^2$ = 0.89, see *Figure 2B,i*).

We next assessed whether cues altered perceptual biases for motion, i.e. whether cues increased the influence of prior beliefs on stimulus classification. Although participants were generally unbiased in their tendency to respond left or right across conditions (choice criterion (c), grand mean F (1, 24) = 1.45, p=0.24), a variance × cue interaction was found such that c was increased on low variance disgust-cued trials, but reduced on high variance disgust-cued trials (V × P interaction, F(1, 24) = 10.46, p=0.004, partial $\eta^2$ = 0.30). Follow-up paired-samples t-tests on this effect revealed that on trials following neutral cues, c did not differ between noise levels (CB NH – NL; t(24) = 0.26, p=0.80), whereas disgust cues increased rightward bias for low variance trials (CB DH – DL t(24) = 3.76, p<0.001). These results demonstrate that unseen, arousing cues selectively increased biases for low noise (high precision) stimuli, in the absence of any differences in the speed or accuracy of motion discrimination.

## Arousing cues reverse Noise-Induced metacognitive uncertainty

To quantify the impact of sensory noise and arousing cues on choice uncertainty, we applied a signal-detection theoretic (SDT) approach to modelling confidence reports (*Galvin et al., 2003*; *Maniscalco and Lau, 2012*). This model yielded M-Ratio and M-Bias parameters, which quantify the objective sensitivity and bias of confidence reports, respectively (*Maniscalco and Lau, 2012*). According to SDT, an M-Ratio (m′/d′) of one indicates optimal metacognitive sensitivity (i.e., confidence ratings exhaust sensory information), with lower ratios indicating poorer metacognition. Alternatively, the M-Bias parameter describes the amount of sensory evidence needed to report a particular level of confidence, with higher values indicating a higher overall subjective uncertainty (i.e., a more conservative confidence bias).

Consistent with interoceptive inference, we found that arousing disgust cues counter-acted the conservative bias induced by high sensory noise (F(1, 24) = 6.19, p=0.020, partial $\eta^2$ = 0.21), see *Figure 2C,iv*. Following neutral cues, confidence reports were significantly more conservative for noisy stimuli (MB NH – NL; t(24) = 2.25, p=0.034), reproducing the previously reported impact of stimulus noise on uncertainty (*Spence et al., 2016*). In contrast, disgust cues reduced this effect, decreasing uncertainty for high variance trials and increasing it for low-variance trials (MB DH – DL; t (24) = −0.197, p=0.85). We also assessed whether these effects were independent of metacognitive sensitivity (i.e., that shifts in uncertainty related to an overall reduction of metacognitive sensitivity), repeating our factorial analysis for M-Ratio. Indeed, cues did not disrupt or alter metacognitive sensitivity; no significant effects were found for M-Ratio (all p>0.6). Additionally, overall M-Ratio and M-Bias did not correlate significantly with one another (r = 0.37, p=0.07). These results demonstrate that perceptual and metacognitive biases for noisy stimuli are selectively altered by arousing disgust cues, even in the absence of performance differences in perception or metacognition.

## Pupillary responses integrate sensory noise and interoceptive arousal

We next determined whether trial-by-trial fluctuations in confidence were related to cardiac or pupillary responses, and if cues successfully altered arousal to modulate these relationships. To do so, we applied a hierarchical general linear modelling approach to estimate the time course of pupillary and cardiac responses, and the encoding of our explanatory variables (e.g., cue valence, sensory noise, confidence and interactions thereof) in these measures. We further performed post-hoc contrasts, or example on the main effect of cue valence or confidence, to delineate the shape of significant interactions. We used a non-parametric, cluster-based permutation t-test (*Hunt et al., 2013*; *Hauser et al., 2015*) to determine when, with respect to trial time, our experimental variables were significantly encoded in evoked physiological responses. This procedure controlled for the family-wise error rate, while simultaneously accounting for variability in trial difficulty, as measured by RT and signal mean (see Materials and methods for more details).

Inspection of the grand mean response for each measure revealed a canonical orientation response locked to trial onset (i.e., the forward mask), as characterized by pupillary dilation (grand

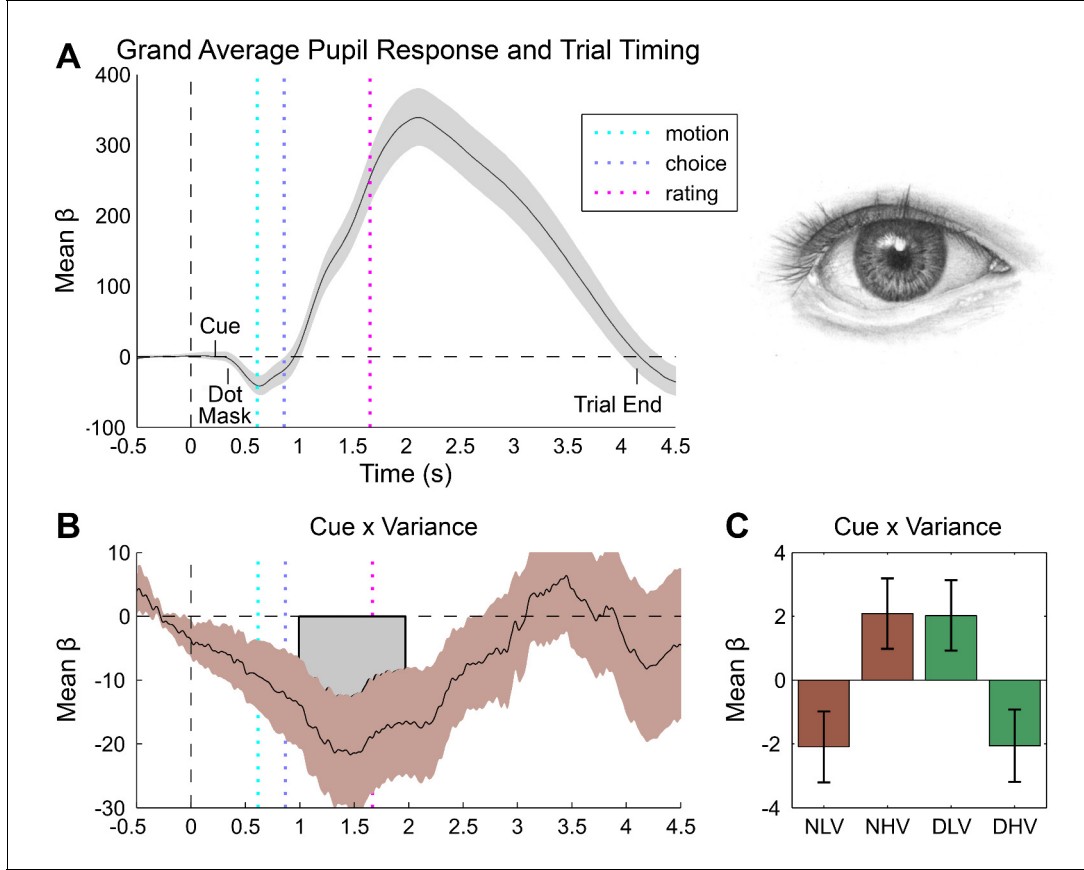

**Figure 3.** Pupillometry results. (**A**) Results of general linear modelling (GLM) of pupil responses; the pupil grand mean response function shows a canonical orientation response, peaking during confidence rating before returning to baseline in the 2–3 s jittered inter-trial interval. (**B**) As predicted, pupillary fluctuations encode the interaction of exteroceptive noise and unexpected internal arousal, time locked to the response interval and onset of confidence rating. (**C**) For illustration, mean response for each condition, extracted from significant time-window controlling for all explanatory and nuisance variables. GLMs were fit across all trials to each time point of the pupil series. Explanatory variables encoded main effects of stimulus noise, variance, confidence, and interactions thereof, revealing the amplitude and timing of each effect. The effects are independent from task-difficulty; trial-wise mean signal and RT were controlled in all analyses. Significance assessed using a cluster-based permutation t-test, cluster alpha = 0.05; cluster shown by shaded grey patch. See Materials and methods for more details.

The following figure supplement is available for figure 3:

**Figure supplement 1.** Additional pupil effects of interest.

mean peak at 2110 ms post-baseline, *Figure 3A*) and heart rate deceleration (grand mean trough at 1900 ms post-baseline, *Figure 4A*) (*Sokolov, 1963*; *Graham and Clifton, 1966*). Consistent with its impact on discrimination difficulty, sensory noise increased pupil dilation (peak effect = 2121 ms post-baseline, duration 554–2377 ms, max β = 24.74, cluster p=0.014) (*Kahneman and Beatty, 1966*; *Murphy et al., 2014b*). Confidence showed a biphasic relationship with dilation depending on trial time, marked by greater dilation during stimulus presentation (peak dilation effect = 712 ms post-baseline (pb), max β = 11.35, Minimum Cluster p=0.038, duration 676–1560 ms post-baseline), but increased constriction during confidence rating (peak constriction effect = 2273 ms post-baseline, max β = −18.53, Minimum Cluster p=0.038, duration 676–1560 ms post-baseline), see *Figure 3—figure supplement 1A*. This effect may reflect distinct neurophysiological contributions from stimulus processing vs post-stimulus evidence accumulation mechanisms (*Pleskac and Busemeyer, 2010*; *Lebreton et al., 2015*; *Lempert et al., 2015*). Confirming that our manipulation successfully modulated arousal, unseen disgust cues significantly increased both pupil dilation and cardiac acceleration (*Figure 3—figure supplement 1B*, and *Figure 4c*), with increased pupil dilation during motion choice (peak at 1596 ms post stimulus, duration 1686–2403 ms, max β = 21.85, cluster p=0.032) and greater cardiac acceleration during confidence ratings (peak effect 3200 ms, duration = 2900–3700 ms, max β = 0.31, cluster p=0.044). Confidence was also related to heart-rate acceleration throughout the trial, with greater confidence linked to a faster heart rate in the interval lasting from stimulus presentation to ratings (peak effects at 500 ms and 3900 ms, durations 100–1000 ms and 1600–4100 ms, max β = 0.31, cluster Ps = 0.046 and 0.002), see *Figure 4A*.

Pupil responses also encoded the interaction of cue and motion variance in the same time interval as the overall cue main effect, with cues reversing the dilatory effect of sensory noise (peak effect 1467 ms post-baseline, duration 1492–2472 ms, min β = −21.67, cluster p=0.034, *Figure 3C*). Crucially, this effect was related to confidence in a positive three-way interaction (peak effect 1512 ms post-baseline, duration 683–2099 ms, max β = 21.22, cluster p=0.008, *Figure 3—figure supplement 1C*), demonstrating that trial-by-trial fluctuations in subjective confidence tracked the cue-induced reversal of pupillary noise encoding. This finding mirrors our primary behavioural effect, indicating that the impact of disgust cue on confidence biases relates to a shift in the mapping between noise-induced uncertainty and physiological responses. In contrast, cardiac signals were insensitive to sensory noise or noise by cue interactions. Instead, the magnitude of the cue-related cardiac main effect negatively interacted with confidence (peak effect 2800 ms post-baseline, duration 2800–3200 ms, min β = −0.30, cluster p=0.044), supporting a reversal in the mapping between heart rate acceleration and subjective uncertainty (*Figure 4C,D*). This latter effect demonstrates that experimentally induced increases in arousal disrupt the typical relationship of heart-rate acceleration and confidence.

## Discussion

Our results demonstrate that unexpected arousal regulates the influence of sensory precision on perceptual uncertainty. This integration of expected internal state and the precision of sensory inputs is consistent with an interoceptive inference mechanism (*Seth, 2013*; *Barrett and Simmons, 2015*), and strongly supports a role for decision-independent sources in guiding confidence. This study thus motivates a revised view of metacognition as incorporating beliefs about both physiological states and the precision of actual sensory inputs.

In general, we demonstrate consistent correlations of trial-by-trial confidence with interoceptive arousal, as indexed by both cardiac acceleration and pupil dilation. In contrast to the linear positive correlation observed for cardiac acceleration, pupil dilation covaried biphasically with subjective confidence, reversing from positive to negative during subjective ratings (see *Figure 3—figure supplement 1A*). This result may partially account for recent findings that distinct stimulus-related and post-decisional computations underlie the representation of confidence (*Lebreton et al., 2015*; *Rahnev et al., 2015*), and corroborates the previously reported link between pupil variability and confidence for an auditory discrimination task (*Lempert et al., 2015*). Furthermore, although interoceptive (i.e., cardiac) sensitivity and meta-cognition for memory have previously been related to one another (*Garfinkel et al., 2013*), our study is the first to show that confidence reports for perception correlate positively with cardiac acceleration. These results thus demonstrate a close link between perceptual confidence and interoceptive arousal, even when accounting for decision difficulty.

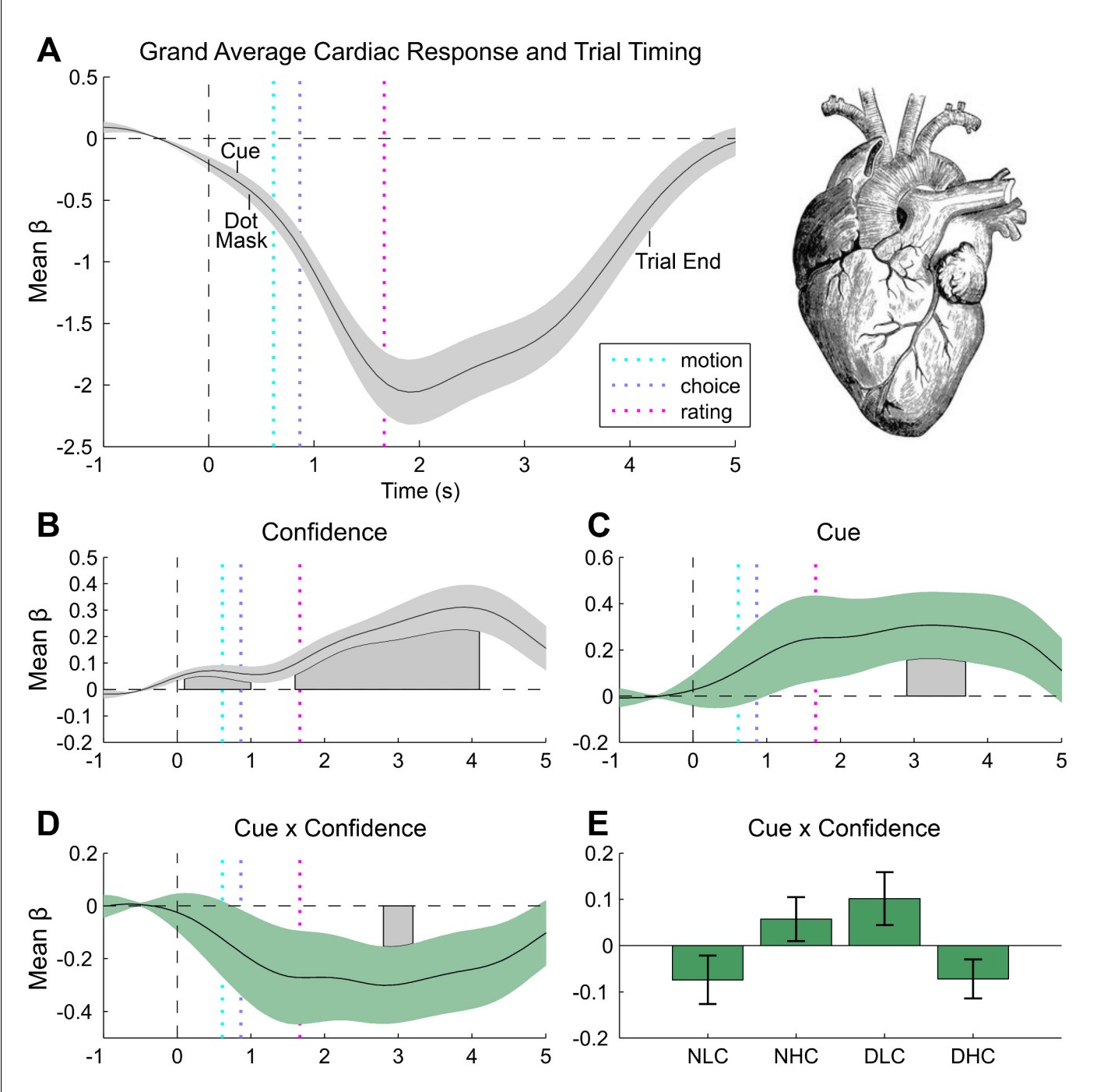

**Figure 4.** Cardiac results. (A) Grand mean cardiac response function showing canonical heart rate deceleration orientation response, and trial timings. (B) Subjective confidence ratings encoded by greater heart rate acceleration, beginning with stimulus onset and peaking during ratings. (C) Unseen disgust cues increase heart rate during confidence rating. (D) This effect interacts with confidence, effectively reversing the mapping of cardiac acceleration and subjective uncertainty. (E) To illustrate this effect, trials were median split into high and low confidence for each disgust condition (e.g., neutral low confidence, NLC), and mean response was extracted from within the significant cue by confidence window. Results of general linear modelling of instantaneous heart rate, with explanatory variables encoding the main effects of stimulus noise, variance, confidence, and interactions thereof, revealing the amplitude and timing of each effect. Effects are independent from task-difficulty; trial-wise mean signal and RT were controlled in all analyses. Significance assessed using a cluster-based permutation t-test, cluster alpha = 0.05; cluster shown by shaded grey patch. See Materials and methods for more details.

However, our use of masked affective cues enabled us to go beyond mere correlation, to assess the causal influence of unexpected increases in pre-stimulus arousal on the perception of sensory noise. Across perceptual, metacognitive, and pupillary measures we observed significant interactions between cue valence and sensory noise. Our disgust cues evoked task-orthogonal, unpredictable increases in both cardiac and pupillary responses. These 'arousal prediction errors' (APEs) counteracted the influence of sensory noise on confidence, supporting the recent proposal that interoceptive inference weights the confidence or precision of exteroceptive sensory signals (*Seth, 2013*; *Barrett and Simmons, 2015*; *Chanes and Barrett, 2016*). This mechanism was also evident in pupillary signals, where the impact of cues on the encoding of sensory noise correlated with trial-by-trial confidence.

Pupil dilation has previously been shown to relate to the overall gain or representational stability of the cortical hierarchy (*Servan-Schreiber et al., 1990*; *Eldar et al., 2013*; *Cheadle et al., 2014*; for review, see *Hauser et al., 2016*). Our results corroborate this proposal, suggesting that pupil variability indexes the impact of unexpected arousal on perceptual precision. In contrast, while cues also shifted the relative mapping of heart rate and confidence, we did not observe an influence of sensory noise on the heart. This may reflect either an issue of causality – our cardiac effects may simply be downstream of cue-induced central nervous system arousal – or it may reflect a more specific encoding of interoceptive but not exteroceptive certainty. Future pharmacological studies using cardiac or noradrenergic-specific blockades will be essential to further tease apart these mechanisms.

Because our manipulation of arousal was by design independent from discrimination accuracy, these results are difficult to accommodate within feed-forward observer models, which posit that confidence depends solely on the information determining stimulus detection (*Galvin et al., 2003*; *Lau and Rosenthal, 2011*; *Maniscalco and Lau, 2012*). However, it is worth considering alternative computational views. For example, one possibility is that arousal alters the overall rate of evidence accumulation or the decision threshold (*Kiani and Shadlen, 2009*; *Vinck et al., 2015*). On this model, arousal would increase confidence by offsetting the overall impact of sensory noise. Similarly, a dynamic or two-stage model could potentially account for decision-independent reductions in confidence (*Pleskac and Busemeyer, 2010*), if arousal linearly shifts the confidence criterion. However, neither model would predict the nonlinear interaction of unexpected arousal and confidence observed here.

Interestingly, we also observed an interaction between cue valence and sensory noise for participant response bias. Although all participants were right handed, disgust cues seemed to enhance a slight rightward bias for high precision stimuli. This result may point to a role for unconscious arousal in strengthening the influence of priors on perceptual inference. Although motion directions were presented randomly across trials, the prevalence for participants to engage in the 'gambler's fallacy', in which a 'streak' of repeated outcomes leads to increased belief that this outcome is more likely, is well documented (*Tversky and Kahneman, 1971*, *1974*). This fallacy constitutes an erroneous belief that one outcome (e.g., leftward motion signals) is more likely than the next. The suggestion here is that by boosting the precision of pre-stimulus beliefs (i.e., expected precision), participants come to believe that stimuli following arousing cues will conform to their (erroneous) motion expectations.

This interpretation is consistent with the more general role of expected precision in bottom-up and top-down attention (*Feldman and Friston, 2010*). Neurobiologically, expected precision (or volatility) is implemented through gain control by neuromodulation, as regulated by insular, cingulate, pulvinar and other limbic areas rich in neuromodulatory neurons (*Friston et al., 2012*; *Moran et al., 2013*; *Schwartenbeck et al., 2015*). Active inference models suggest that the regulation of expected precision is a central mechanism underlying both bottom-up (i.e., salience) and top-down attention (*Feldman and Friston, 2010*; *Moran et al., 2013*; *Kanai et al., 2015*) Thus, the sudden increase in arousal elicited by cues may correspond to an inflation of expected precision, which would reduce the salience of sensory (exteroceptive) gain in perceptual inference.

However, here we do not explicitly manipulate the underlying probability of receiving an arousing cue. To conclusively determine how the interaction of arousal prediction and expected precision shapes confidence, future work should explicitly manipulate the volatility of interoceptive fluctuations by altering the underlying probability of an arousal change point (*Behrens et al., 2007*; *Summerfield et al., 2011*). Additionally, although here we manipulate arousal and observe correlated changes in cardiac signals and confidence, the causal link to interoception must be established in future investigations, in which cardiac signals are directly manipulated independently of the

central nervous system. This approach, coupled with hierarchical psycho-physiological computational modelling (*de Berker et al., 2016*), could further reveal the interoceptive computations underlying perceptual confidence.

## Conclusions and clinical implications

In the present study, we demonstrate a close linkage of perceptual confidence, unexpected arousal, and related interoceptive signals. Across perceptual, physiological, and subjective measures we demonstrate that the encoding of sensory noise is weighted by interoceptive arousal. These results may have important implications for understanding medical and psychiatric disorders, in which patients exhibit chronic alterations in arousal or interoception. Substance abuse, psychosis, anxiety, and depression for example have been linked to altered heart-rate variability, physiological responses, and interoceptive sensitivity (*Dawson et al., 1977*; *Hoehn-Saric and McLeod, 2000*). Our results suggest that the altered psychophysiology of these patients may cause them to perceive an unrealistically (un)-certain world.

# Materials and methods

## Participants

29 participants took part in the experiment at University College London (UCL). Previous studies examining the impact of sensory noise on confidence (*Zylberberg et al., 2014*; *Spence et al., 2016*) and pupillometric responses during decision making (*Murphy et al., 2014b*) have reported samples of 7–20 participants. To ensure a robust estimate of our behavioural and physiological effect while accounting for potential missing data (due to e.g., trials rejected due to blinks), we recruited a larger sample of 29 participants (17 F) aged 18–39 (M = 25.4, SD = 5.0) in total.

All participants had normal or corrected-to-normal vision with no history of neurological or psychiatric disorders. Participants received monetary compensation (£15) for completing the experiment. Informed consent was obtained from all participants, and all procedures were conducted in accordance with the Declaration of Helsinki and with approval from the UCL Research Ethics Committee.

## Experimental setup

### Overview

Participants completed 10 blocks of a psychophysical metacognition task consisting of 640 trials divided evenly between four conditions, with enforced breaks following each 64 trial block. The task required participants to judge the average direction of a global motion signal and to make confidence ratings on each trial, following a masked interoceptive cue (see Trial Structure, below for more details). Physiological signals (ECG and pupillometry) were monitored throughout the experiment. At the conclusion of the experiment, participants were, (1) asked if they had noticed anything unusual about the presented faces, (2), debriefed that on half the trials there had been a hidden emotional face, and (3) completed 200 trials of a forced-choice cue-identity detection task to quantify masking efficacy. Pupil signals were synchronized with stimulus timing using the Eyelink Toolbox (*Cornelissen et al., 2002*). Cardiac signals were amplified using an Asalab System and recorded with Visor2 2.0 software (ANT Neuro Recording), synchronised via a parallel port trigger from the stimulus PC.

### Participant instruction and training

Participants were instructed that the purpose of the task was to assess their ability to discriminate an average motion signal relative to vertical, and also to make introspective confidence ratings about the accuracy of such choices. During the briefing and electrode placement, participants were instructed that they must remain as still as possible and maintain central fixation at all times during the experiment, in order to limit recording artefacts. With respect to the neutral face mask, participants were instructed to maintain central fixation and to otherwise ignore the facial stimulus as it merely cued trial onset. All participants completed a brief training protocol prior to the main task consisting of 30 motion discrimination trials without confidence ratings, followed by 10–15 trials practicing both stimulus discrimination and confidence ratings. During training choice feedback was provided by altering the colour of the fixation dot to indicate correct/incorrect responses. Finally,

following previous studies in this area, participants were instructed to reflect carefully on their subjective confidence on each trial and to generally make use of the entire confidence scale (*Fleming et al., 2010*; *Maniscalco and Lau, 2012*; *Lempert et al., 2015*). All participants indicated complete understanding of the task before proceeding to the main experiment.

## Stimuli, trial structure, and experimental design

To manipulate both sensory precision and arousal, we developed a disgust-cued psychophysiological metacognition task, in which the masked presentation of disgusted faces cued trial onset following a variable 2–3 s inter-trial interval (ITI).

Motion stimuli comprised a random dot stimulus moving in an approximately upward-vertical direction. The stimuli were always presented at 100% coherence, that is, all dots were "signal dots" but the distribution of motion vectors was varied parametrically. To control discrimination performance, the mean signal (the average direction of dots) was controlled across trials using an adaptive 2 up 1down staircase which converges on 70.7% accuracy (*Cornsweet, 1962*; *Fleming and Lau, 2014*). To ensure orthogonal manipulation of signal mean and variance, responses to high variance trials were not included in the staircase. Instead, on each high variance trial mean orientation was generated using the signal mean from the previous low variance trial of the same cue condition. Disgust and neutral trials were thresholded separately to allow for the possibility of the priming condition contributing to detection accuracy (although overall they did not).

Sensory noise was manipulated independently by adjusting the standard deviation of the mean direction across conditions. 1000 black dots of radius 0.08 degrees visual angle (DVA) were presented for 250 ms within a 15.69 DVA diameter circular array at random starting positions, with dots advancing 0.06 DVA per frame. Dots which moved beyond the stimulus aperture were replaced by dots at the opposite edge to maintain constant dot density. To prevent fixation on the local motion directions, all dots had a randomized limited lifetime of maximum 93% (14 frames). On each trial the motion signal was thus calculated using the formula:

$$\text{Dot Directions} = (\text{Left vs Right}) * \text{Mean Orientation} + \text{Gaussian Noise} * \text{Standard Deviation}$$

The experimental paradigm thus consisted of a within-subject 2 x 2 factorial design manipulating the valence of masked cues (disgusting, neutral) and the variance of the presented motion signal (25° vs 15° SD), resulting in four conditions i.e. disgust high variance (DH), disgust low variance (DL), neutral high variance (NH), and neutral low variance (NL) which were randomly interleaved within each block of trials. For emotional face cues, four paired male faces showing forward-directed disgust or neutral expressions were selected from the Karolinska Directed Emotional Faces database (KDEF; *Lundqvist et al., 1998*) Disgusted faces were selected based on highest mean arousal and intensity scores (*Goeleven et al., 2008*). The original images were manipulated so that only the face was visible, removing any background and hairline. The images were then cropped to 4.90 × 2.41 DVA (height × width) elliptical shapes with a 0.16 DVA Gaussian blur frame. A small amount of blur was also added to obscure visible teeth, a salient feature which can lead to masking failure. All images were centrally presented with a fixation dot at the apex of the nose.

Following established protocols, we used a combination of forward and backwards masking to ensure cues were not consciously visible (*Bachmann and Francis, 2013*; *Overgaard and Overgaard, 2015*). Neutral face identity was pseudo-randomly selected from the stimulus pool to ensure that face identity always changed from cue to mask. A forward mask was created by phase-scrambling all stimulus faces. Each trial began with the presentation of a phase-scrambled face (250 ms duration), followed by a single cue frame (~16.667 ms), and a neutral-face mask (100 ms). Immediately following the neutral face a stationary dot display appeared (250 ms) prior to motion onset (250 ms), followed by another stationary isoluminant dot-mask, which participants viewed while making their perceptual choice (800 ms) and confidence rating (2500 ms). At the end of the perceptual choice interval, a sliding scale marked at four equal intervals by vertical lines appeared, centered within the dot mask. To limit eye movements, the width of the rating scale was restricted to one half the dot-mask radius. All stimuli were presented behind a central fixation dot. Finally, after the rating interval the scale vanished and participants centrally fixated on the dot mask for a 2–3 s randomly jittered ITI. In a separate pilot experiment with an identical set-up, trial timings were verified to be accurate within a millisecond using a photodiode and oscilloscope.

Confidence was rated by moving a small triangular slider along the confidence line using the left and right arrow keys, and recorded as a 0–100 integer. To prevent response preparation the starting point of the slider on each trial was randomly jittered up to 12% to the left or right of the scale mid-point on every trial (*Fleming et al., 2012*; *Fleming and Lau, 2014*). Stimulus delivery and timing were controlled using Psychtoolbox-3 (https://www.psychtoolbox.org/) implemented in MATLAB R2014a (Mathworks Inc., USA). See *Figure 1A* for an illustration of our trial design.

## Luminance control and response timing

As luminance is a primary driver of the pupillary response, we implemented a rigorous procedure to ensure equal mean luminance between conditions and to minimize frame-to-frame luminance changes. The monitor was calibrated with a Minolta CS-100A photometer and linearized in software, giving a mean and maximum luminance of 42.5 and 85 cd/m$^2$, respectively. All stimuli were presented on a grey background at mean luminance. Face stimuli were set to grayscale and pre-processed using the SHINE toolbox in MATLAB, which uses a histogram-matching procedure to balance images both in terms of average luminance and local statistical properties (*Willenbockel et al., 2010*). To minimize luminance changes from face to dot presentation and hence maximize our signal to noise ratio, face stimuli were set to half contrast and altered to match the average luminance of the dot display. Following image pre-processing, all presented stimuli were measured using the photometer positioned at the point of head fixation to ensure equivalent emitted luminance between and within trials.

To stabilize pupil signals across trials, we also adapted our stimuli timings, ITIs, and use of isoluminant dot masks on the basis of a prior dot motion pupillometry study, ensuring a minimum 6 s inter-response interval (IRI) to allow pupil recovery (*Murphy et al., 2014b*). We further stabilized IRIs using response timing, with participants instructed to make their response as accurately as possible within a restricted time window (0–800 ms post motion cessation). On any trial in which the participant exceeded this limit, a red alert text stating 'Too Slow' appeared for 200 ms followed by the usual ITI. Missed trials were excluded from analysis. A pilot study confirmed our 6.166–7.66 s IRI was sufficiently for dilation to return to baseline before the start of the next trial (see *Figure 3A* for global pupil response plotted over trials).

## Physiological monitoring

The experiment took place in an electrically shielded room designed for electroencephalographic recording at the Wellcome Trust Centre for Neuroimaging, UCL. Participant head position was held constant throughout the experiment using a headrest positioned 62 cm from the screen. ECG signals were measured using disposable Ag/AgCl bipolar surface electrodes (100 Foam, Covidien) affixed just below the left and right clavicle and a ground electrode affixed to the nape of the neck using medical tape and Spectra 360 salt-free electrode gel (Parker Laboratories). Prior to electrode placement each contact site was thoroughly cleaned using an alcohol swab. ECG signals were amplified using an Asalab System (ANT Neuro Recording) and recorded via Visor2 2.0 software at a 1024 Hz sampling rate. Changes in pupil diameter were monitored using an Eyelink 1000 eye tracker (SR research) recording at 1000 Hz sampling rate, and synchronized to the stimulus PC using the Eyelink Toolbox, for PsychToolbox (RRID:SCR_002881) (*Cornelissen et al., 2002*). At the start of the experiment the eye tracker was calibrated and validated for each participant's right eye using an automated 9-point tracking test.

## Post-task masking efficacy measure

To empirically validate the efficacy of our masking procedure, at the end of the task participants completed a forced-choice valence identification task. This involved 200 trials of identical set-up to the main experiment, but with a fixed 1 s ITI. Participants were instructed that during the main experiment there had occasionally been an emotional face presented just before the neutral mask, and that they were to now try to detect on every trial whether the "hidden face was emotional or neutral". Participants were encouraged to make their first choice even if they were unsure.

## Analysis

### Behavioural

Prior to analysis, orientation staircases, detection thresholds, and confidence histograms were plotted for each participant. In three participants, thresholds failed to stabilize resulting in extreme (>3 SD) median signal orientations. A fourth participant exhibited extreme confidence behaviour (> 75% of trials marked 100% confidence), resulting in four total exclusions from behavioural analyses, final N = 25. To establish the efficacy of our cue masking, we calculated D-prime for sensitivity to discriminate positive vs neutral valence for each of the four face identities. Mask efficacy was determined by calculating the 95% confidence intervals for each sensitivity measure.

To allow staircase stabilization, the first 25% of trials (i.e. the first two blocks) were excluded from behavioural analyses, as well as any missed trials or trials with outlier RTs (i.e. absolute z-scored RT > 3). As verification of this procedure, we analysed accuracy over each block in a one-way ANOVA (factor: block); no significant effect of block on detection accuracy was found when excluding the initial two blocks, confirming task stability. To facilitate signal-theoretic modelling, confidence ratings were binned into four quartiles (*Maniscalco and Lau, 2012*; *Fleming and Lau, 2014*). Type-I performance measures median reaction time, median signal orientation, d-prime, and criterion were calculated for each condition and participant.

Metacognitive behaviour was analysed using a signal-detection theoretic (SDT) Meta-d' modelling approach to estimate objective confidence sensitivity and bias (*Maniscalco and Lau, 2012*). This approach quantifies an individual's metacognitive ability by comparing the sensitivity of their subjective confidence ratings (e.g., the probability high confidence | correct response vs high confidence | error response) across trials to the expected performance of an optimal observer (under SDT assumptions) given their actual discrimination performance. By comparing the actual metacognitive sensitivity to expected (e.g. M-prime/D-prime), the Meta-d' model quantifies an individual's introspective sensitivity and bias while controlling for the confounding impact of type-I performance (see *Maniscalco and Lau, 2012* for a full methodological treatment). The type-II measures M-Ratio (MR), and M-Bias (MB, or Meta-Criterion), which characterize metacognitive sensitivity and bias (i.e., uncertainty) respectively, were thus calculated using maximum likelihood estimation implemented in freely available MATLAB (Mathworks Inc, version R2014a) code separately for each condition (http://www.columbia.edu/~bsm2105/type2sdt/). All type-I and II measures were entered into $2 \times 2$ repeated measures ANOVAs with factors cue valence (disgust, neutral) and variance (high, low), $\alpha = 0.05$. All ANOVAs and associated t-tests were conducted in JASP (version 0.7.1).

### Physiological Pre-processing

Pupil data were imported and pre-processed in MATLAB using the Fieldtrip package (RRID:SCR_004849, *Oostenveld et al., 2010*). Data for each condition were epoched according to the onset of the forward mask from −500 ms baseline to 4166 ms (rating offset), before applying automatic blink detection and linear interpolation, blink rejection, linear de-trending, low pass-filtering, and a combination of manual and automatic artefact rejection. Blinks were detected as any sample in which amplitude dropped below 600 arbitrary units dilation and were linearly interpolated, such that if any trial began or ended with a blink the interpolation was based on the first reliable sample. Any trial where more than 25% of samples were marked as blinks were rejected from the analysis. All trials were then linearly detrended and low-pass filtered at a 30 Hz cut-off, manually inspected for artefacts and passed through a final automatic artefact detection, searching for the unreliable pupil lock based upon the absolute maximum of the trial's first derivative, with any trial beyond 3 SD rejected. These procedures resulted in a mean rejection rate of 19.30% (SD = 9.33) of trials across participants. Pupil measures could not be obtained from two participants due to technical difficulties at recording, and one additional participant was rejected due to excessive blink artefacts, resulting in a final N = 26.

ECG data were imported to MATLAB using custom code adapted from FieldTrip, epoched for each condition according to the onset of the forward mask from −1000 ms baseline to 4166 ms (rating offset). QRS complex detection was implemented on the raw data after downsampling to 200 Hz with the Pan-Tomkins algorithm (*Pan and Tompkins, 1985*) and supplemented by manual inspection and editing. Data were flagged for possible artefacts by splitting the raw series into segments (30 s segment length), and any segment with average heart rate outside the 50–120 beats per minute

(bpm) or any individual interbeat-interval (IBI) between 0.45–1.40 s was marked for manual inspection. Artefactual samples were then manually marked and removed from analyses, resulting in a 6.9% (SD = 7.8) mean rejection rate. Instantaneous heart rate was calculated using the IBI and converted to a continuous time series by interpolating the heart rate (using spline interpolation), upsampling to a 10 Hz sampling rate.

### Physiological timeseries analysis

Following pre-processing, pupillary and ECG time series were analysed using a hierarchical general linear modelling approach. To do so, each trial was first baseline corrected for the pre-stimulus interval. Design matrices were then constructed with explanatory regressors encoding the main effect of stimulus variance, cue valence, and confidence. Additionally, we modelled the interaction of confidence with variance and cue valence, the cue by variance interaction, and the three-way cue × variance × confidence interaction. Finally the model included the mean orientation signal and RT for each trial, to control for possible confounding effects of detection difficulty. Thus, for each physiological measure and at each sampled time bin, we fit a regression model of the form:

$$y^{pupil|hr} = \beta_0 + \beta_1 x^{Noise} + \beta_2 x^{Cue} + \beta_3 x^{Confidence} + \beta_4 x^{Noise*Confidence} + \beta_5 x^{Cue*Confidence} + \beta_6 x^{Cue*Noise*Confidence} + \beta_7 x^{Mean} + \beta_8 x^{RT} + \varepsilon$$

Where x denotes the normalized vector of the respective independent variable across all trials independently for each participant, β is the effect size, and ε is measurement noise. Using a summary statistic approach, we tested the consistency of the individual time series at the group level conducting t-tests for the positive and negative effect of each regressor, corrected for multiple comparisons using a standard cluster-based permutation test (p <0.05 cluster-extent correction, n = 500 permutations, height threshold t = 2) (*Hayasaka et al., 2004*; *Hunt et al., 2013*; *Hauser et al., 2015*) using custom code in MATLAB. This approach allowed us to assess when a particular condition or interaction of interest was significantly encoded in that variable, maximizing temporal sensitivity without the assumptions of a deconvolution approach (*Wierda et al., 2012*).

## Acknowledgements

The authors thank Chris Frith, Karl Friston, Steve Fleming, Jonathan Smallwood, and John Greenwood for insightful comments and input on the design, analysis, apparatus, and manuscript. This work was supported by the Wellcome Trust (grant 100227 [MA, GR], grant 095939 [JSW]), an ERC Starting Grant to DSS (310829), and a Swiss National Science Foundation Grant to TUH (151641). The Wellcome Trust Centre for Neuroimaging is supported by core funding from the Wellcome Trust (091593).

## Additional information

### Funding

| Funder | Grant reference number | Author |
|---|---|---|
| Wellcome Trust | 100227 | Micah Allen<br>Geraint Rees |
| European Research Council | 310829 | D Samuel Schwarzkopf |
| Wellcome Trust | 095939 | Joel S Winston |
| Wellcome Trust | 091593 | Micah Allen<br>Darya Frank<br>Joel S Winston<br>Tobias U Hauser<br>Geraint Rees |
| Schweizerischer Nationalfonds zur Förderung der Wissenschaftlichen Forschung | 151641 | Tobias U Hauser |

The funders had no role in study design, data collection and interpretation, or the decision to submit the work for publication.

## Author contributions

MA, Conception and design, Acquisition of data, Analysis and interpretation of data, Drafting or revising the article; DF, Conception and design, Acquisition of data, Drafting or revising the article; DSS, Conception and design, Drafting or revising the article, Contributed unpublished essential data or reagents; FF, Analysis and interpretation of data, Drafting or revising the article, Contributed unpublished essential data or reagents; JSW, TUH, GR, Conception and design, Analysis and interpretation of data, Drafting or revising the article

## Author ORCIDs

Micah Allen, http://orcid.org/0000-0001-9399-4179
Darya Frank, http://orcid.org/0000-0001-6081-6755
Francesca Fardo, http://orcid.org/0000-0002-9974-6261
Joel S Winston, http://orcid.org/0000-0002-3957-0612
Tobias U Hauser, http://orcid.org/0000-0002-7997-8137

## Ethics

Human subjects: All participants provided informed consent, and all procedures were conducted in accordance with the Declaration of Helsinki and with approval from the UCL Research Ethics Committee (UCL Ethics Project ID Number: 4223/002). All collected data are subject to the approved data protection practices (Z6364106/212/09/10).

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
