## [Decision Letter]

Thank you for submitting your article "Perceptual confidence integrates interoceptive arousal and sensory noise" for consideration by *eLife*. Your article has been favorably evaluated by Sabine Kastner (Senior Editor) and three reviewers, one of whom, Haozhe Shan (Reviewer #1), is a member of our Board of Reviewing Editors. The following individuals involved in review of your submission have agreed to reveal their identity: Michael Breakspear (Reviewer #2) and Philip Corlett (Reviewer #3).

In "Perceptual confidence integrates interoceptive arousal and sensory noise", Dr. Allen and colleagues presented a carefully designed experiment using psychophysical and psychophysiological methods. Using subconscious presentation of affectively salient stimuli, the authors manipulated the arousal of the subjects while they judged their perceptual confidence. The findings, showing that physiological arousal, as reflected by changes in pupil sizes and heart rate, influences judgment of perceptual confidence, adds to our understanding of how interoception of arousal influences seemingly non-affective perceptual processes. However, the reviewers have concerns about several aspects of the paper. They should be addressed with major revisions.

The biggest concern that the reviewers have is that the interpretations and theoretical implications do not seem to be well grounded in evidence. The concept of "gain", in either "cortical gain" or "sensory gain", was used for a few times to connect the findings to research in systems/computational neuroscience. However, what does cortical gain and sensory gain mean exactly in these contexts, in what fashion are they modulated by the experimental manipulations, and how exactly they produce the effects that the authors attribute to them, are unclear. "Gain" is a term that simply describes changes in the power of signals. It does not have the non-linearity or other complex properties that the authors seemed to be using to explain their findings.

In addition, the author invoked the adaptive gain theory, which states that arousal boosts stimulus-related presentation in the cortex and reduces noise. How this mechanism would produce a non-linearity is not clear, and it does not seem to be compatible with the authors' finding that perceptual performance is not enhanced by physiological arousal. The necessity for something complex, i.e. the non-linear hypothesis illustrated in Figure 1, is questionable. The summation hypothesis (2) is sufficiently addressed. The authors described it in the text as the summation of "sensory and physiological arousal", while describing it in Figure 1 as a summation of "uncertainty from extraceptive and interoceptive channels". Why arousal equates uncertainty, and why arousal cannot boost confidence by linearly offsetting the effects of noise are unclear. The connection to predictive coding – the only relevance of predictive coding here is that, as the authors have stated, predictive coding goes up a hierarchical structure and gets multi-modal at a higher level. It seems like there's nothing from the current findings that is related to the core concepts of predictive coding – expectations and error representations. Finally, the reviewers had concerns about the connection to locus coeruleus and particular neurotransmitters. We do not have an understanding of the locus coeruleus that allows us to directly link pupil dilation to a certain neurotransmitter etc. The reviewers encourage the authors to take out or reduce discussions of these connections. They may be more appropriate for the Discussion.

Secondly, it is important to note that what is being manipulated in the study appears to be physiological arousal, not heart rate or pupil dilation. Changes in heart rate and pupil dilation are correlated with changes in arousal, but they themselves are not directly manipulated. Therefore, the connection between interoception of heart rate and pupil dilation and confidence is correlational. The causal relationship that the results seem to support is one between arousal and confidence, not heart rate/pupil and confidence. The reviewers advise the authors to change the wording and instead focus the conclusions on arousal, rather than interoception of heart rate or pupil dilation.

Finally, in terms of statistics, the authors need to address the family-wise error rate across the study. The authors should state how many independent analyses were performed on the data (i.e. multiple ANOVAs) and how the authors corrected the family-wise error rate caused by that.

---

## [Author Response]

*In "Perceptual confidence integrates interoceptive arousal and sensory noise", Dr. Allen and colleagues presented a carefully designed experiment using psychophysical and psychophysiological methods. Using subconscious presentation of affectively salient stimuli, the authors manipulated the arousal of the subjects while they judged their perceptual confidence. The findings, showing that physiological arousal, as reflected by changes in pupil sizes and heart rate, influences judgment of perceptual confidence, adds to our understanding of how interoception of arousal influences seemingly non-affective perceptual processes. However, the reviewers have concerns about several aspects of the paper. They should be addressed with major revisions.*

Thank you for you the thorough assessment of our paper, as well as the positive and encouraging feedback on our work. We have now made substantial revisions, including a completely revised Introduction and Discussion to make our study and the interpretation more clear. We also revised the Methods and Results section to clarify our procedures. We hope that our revisions will successfully address all raised issues.

*The biggest concern that the reviewers have is that the interpretations and theoretical implications do not seem to be well grounded in evidence. The concept of "gain", in either "cortical gain" or "sensory gain", was used for a few times to connect the findings to research in systems/computational neuroscience. However, what does cortical gain and sensory gain mean exactly in these contexts, in what fashion are they modulated by the experimental manipulations, and how exactly they produce the effects that the authors attribute to them, are unclear. "Gain" is a term that simply describes changes in the power of signals. It does not have the non-linearity or other complex properties that the authors seemed to be using to explain their findings.*

*In addition, the author invoked the adaptive gain theory, which states that arousal boosts stimulus-related presentation in the cortex and reduces noise. How this mechanism would produce a non-linearity is not clear, and it does not seem to be compatible with the authors' finding that perceptual performance is not enhanced by physiological arousal. The necessity for something complex, i.e. the non-linear hypothesis illustrated in Figure 1, is questionable. The summation hypothesis (2) is sufficiently addressed. The authors described it in the text as the summation of "sensory and physiological arousal", while describing it in Figure 1 as a summation of "uncertainty from extraceptive and interoceptive channels". Why arousal equates uncertainty, and why arousal cannot boost confidence by linearly offsetting the effects of noise are unclear. The connection to predictive coding – the only relevance of predictive coding here is that, as the authors have stated, predictive coding goes up a hierarchical structure and gets multi-modal at a higher level. It seems like there's nothing from the current findings that is related to the core concepts of predictive coding – expectations and error representations. Finally, the reviewers had concerns about the connection to locus coeruleus and particular neurotransmitters. We do not have an understanding of the locus coeruleus that allows us to directly link pupil dilation to a certain neurotransmitter etc. The reviewers encourage the authors to take out or reduce discussions of these connections. They may be more appropriate for the Discussion.*

We apologize that our original framing was unclear. We have now made a full revision of our Introduction and Discussion, to clarify our hypotheses and their relationship to our experimental design. Additionally, we simplified the terminology to make it more understandable for the broad audience of e*Life*, including a more clear explanation of neural gain as requested. As suggested, we also removed Figure 1 to avoid confusion, and also removed any reference to the locus coeruleus system.

Indeed, as the reviewers suggest, gain is a basic concept from signal processing. While there are interesting links between (local) neural gain, (global) adaptive gain regulation by arousal, and the predictive coding of precision, we agree that these are beyond the scope of our investigation. Our goal with this experiment was to test one specific hypothesis from the recent theoretical literature on “interoceptive inference” and predictive coding; i.e., that unexpected changes in endogenous states (e.g., arousal, heart rate) should reduce the impact of sensory noise (or precision) on confidence (Seth, 2013; Barrett and Simmons, 2015; Chanes and Barrett, 2016), producing a interaction between these two factors. This was implemented in our experimental design by manipulating sensory precision (independently of mean), and presenting subconscious, arousing cues just prior to motion stimuli to create an ‘arousal prediction error’.

In our completely revised Introduction, we now focus on explaining in detail both the ideal observer and interoceptive inference models of perceptual confidence, which motivated our original hypotheses. With respect to the ideal observer model, we now state in our Introduction:

“Computationally, confidence is typically described as the output of a feed-forward ideal statistical observer monitoring sensory (or decision) evidence. […] In both cases, confidence is generated by the bottom-up read-out of sensory information relative to a decision variable, and is assumed to depend on the same information underlying the accuracy of the perceptual decision itself. “

The key thing here is that ideal observer models explicitly rules out accuracy-independent effects on confidence (for review, see Lau and Rosenthal, 2011). This speaks to an important point raised by our reviewers regarding a lack of differences in perceptual accuracy; this is exactly as intended by our experimental design, and is what enables us to rule out a simplistic observer model. In contrast, if confidence reflects interoceptive inference, then we can expect that arousal can affect confidence even if perceptual performance is identical. We now carefully motivate this view in our Introduction as follows:

“While substantial evidence supports the integration of interoceptive arousal and sensory information, these observations are difficult to reconcile with ideal observer models. […] Predictive coding thus hypothesizes that the weight given to sensory noise depends upon expected arousal (Gu et al., 2013; Seth, 2013; Barrett and Simmons, 2015).“

We now simplify the explanation of our hypotheses to contrast the interoceptive inference account described above, versus the optimal observer, which argues that detection-independent arousal should have no effect on confidence. We originally suggested the uncertainty summation hypothesis because 1) interoceptive responses and arousal have previously been linked to decision uncertainty and 2) as cue-induced arousal changes were (by design) orthogonal to detection accuracy, they constitute noise from the point of view of optimal decision-making. However, a strictly feed-forward optimal observer model would still preclude such effects, as they would also impact detection accuracy. One could potentially explain task-independent effects using a two-stage accumulation or detection model, but here arousal would be expected to only increase or decrease confidence. We further revised our Discussion to better explain our results in light of these considerations, which now focuses on:

1) The specific relationship of our findings to previous work on physiology and confidence:

“In general, we demonstrate consistent correlations of trial-by-trial confidence with interoceptive arousal, as indexed by both cardiac acceleration and pupil dilation. […] These results thus demonstrate a close link between perceptual confidence and interoceptive arousal, even when accounting for decision difficulty.”

2) The specific predictions of interoceptive inference vs the standard ideal observer model, and also possible alternative hypothesis (including the interesting idea of a linear offset suggested by the reviewer):

“Because our manipulation of arousal was by design independent from discrimination accuracy, these results are difficult to accommodate within feed-forward observer models, which posit that confidence depends solely on the information determining stimulus detection (Galvin et al., 2003; Lau and Rosenthal, 2011; Maniscalco and Lau, 2012). However, it is worth considering alternative computational views. […] Similarly, a dynamic or two-stage model could potentially account for decision-independent reductions in confidence (Pleskac and Busemeyer, 2010), if arousal linearly shifts the confidence criterion. However, neither model would predict the nonlinear interaction of unexpected arousal and confidence observed here.”

3) A revised and expanded consideration of the response bias:

“Interestingly, we also observed an interaction between cue valence and sensory noise for participant response bias. […] This fallacy constitutes an erroneous belief that one outcome (e.g., leftward motion signals) is more likely than the next. The suggestion here is that by boosting the precision of pre-stimulus beliefs (i.e., expected precision), participants come to believe that stimuli following arousing cues will conform to their (erroneous) motion expectations.”

4) And finally, a consideration of our results from the point of view of expected precision:

“This interpretation is consistent with the more general role of expected precision in bottom-up and top-down attention (Feldman and Friston, 2010). […] This approach, coupled with hierarchical psycho-physiological computational models (de Berker et al., 2016), could more conclusively reveal the interoceptive computations underlying perceptual confidence.”

*Secondly, it is important to note that what is being manipulated in the study appears to be physiological arousal, not heart rate or pupil dilation. Changes in heart rate and pupil dilation are correlated with changes in arousal, but they themselves are not directly manipulated. Therefore, the connection between interoception of heart rate and pupil dilation and confidence is correlational. The causal relationship that the results seem to support is one between arousal and confidence, not heart rate/pupil and confidence. The reviewers advise the authors to change the wording and instead focus the conclusions on arousal, rather than interoception of heart rate or pupil dilation.*

We agree on this important point. Our cue-based paradigm manipulates pre-stimulus arousal. As such, any changes in interoceptive signals (e.g., cardiac acceleration) may be causally down-stream and/or epiphenomenal to our confidence and arousal effects. Although it is interesting to consider the possibility of ‘circular causality’, i.e., a role for the brain in both regulating and being regulated by interoception, this lies beyond the explanatory scope of our paradigm. We revised our Discussion, as advised, to highlight this issue:

“In contrast, while cue-induced acceleration of cardiac signals also shifted their mapping relative to confidence, we did not observe any influence of sensory noise on the heart. This may reflect either an issue of causality – our cardiac effects may simply be downstream of cue-induced central nervous system arousal – or may reflect a more specific encoding of interoceptive but not exteroceptive certainty.”

We also now note in our Discussion, some limitations of the study that should be addressed in future research:

“However, here we do not explicitly manipulate the underlying probability of receiving an arousing cue. […] This approach, coupled with hierarchical psycho-physiological computational modelling (de Berker et al., 2016), could more conclusively reveal the interoceptive computations underlying perceptual confidence.”

*Finally, in terms of statistics, the authors need to address the family-wise error rate across the study. The authors should state how many independent analyses were performed on the data (i.e. multiple ANOVAs) and how the authors corrected the family-wise error rate caused by that.*

Thank you for pointing out this issue; in our original manuscript it was unclear which analyses were a priori and which were post-hoc control analyses. To be clear, this experiment is motivated by one central hypothesis, namely, that unexpected arousal should modulate the impact of sensory noise in confidence (i.e., a nonlinear interaction). This is reflected in our analyses of the metacognitive bias parameters (M-Bias), with the key test being the cue by noise interaction.

We also conducted a variety of control analyses to assess the robustness of our thresholding procedure; i.e., to verify that perceptual sensitivity (D-prime), motion thresholds, reaction times, and block-by-block accuracy did not differ between cue conditions. Additionally, we report that there is no difference in metacognitive sensitivity (m-ratio), as this strengthens the interpretation of the uncertainty measure as being performance independent. As we now explain in our revised Introduction, the stability of these measures increases the contrast between our results and the ideal observer model, and rules out a variety of trivial confounds.

“Computationally, confidence is typically described as the output of a feed-forward ideal statistical observer monitoring sensory (or decision) evidence. […] In both cases, confidence is generated by the bottom-up read-out of sensory information relative to a decision variable, and is assumed to depend on the same information underlying the accuracy of the perceptual decision itself.”

For completeness, we also performed an exploratory analysis of perceptual bias. We performed this analysis because 1) our experimental design precluded any changes in motion sensitivity and 2) changes in bias in the absence of a sensitivity difference potentially highlight a difference of conscious (subjective) motion perception (see Peters et al., 2016). While this result is more speculative, we believe that it is important because it shows that our predicted cue by noise interaction is present across perceptual, metacognitive and physiological measures. We now include a new section of our Discussion considering this result:

“Interestingly, we also observed an interaction between cue valence and sensory noise for participant response bias. […] The suggestion here is that by boosting the precision of pre-stimulus beliefs (i.e., expected precision), participants come to believe that stimuli following arousing cues will conform to their (erroneous) motion expectations.”

Finally, we note that our two behavioral interactions (M-ias and Choice Bias) both survive Bonferroni correction for two comparisons (minimum p < 0.020); more importantly for the purposes of reproducibility, both can be considered large effect sizes (η2 for m-ias interaction = 0.21; bias = 0.30) by the metric proposed by Cohen (1988), which describes small (η2 = 0.01), medium (η2 = 0.06), and large (η2 = 0.14) effects. We have now carefully revised our methods section to more clearly state which analyses are a priori vs. post-hoc (as requested), and also emphasize the cross-measure consistency of the interaction in our Discussion:

“In a series of control analyses, we confirmed that 1) staircases were stable across trials and between conditions, 2) staircases successfully controlled for potential cue-induced differences in detection difficulty, and 3) the masking procedure successfully prevented the detection of cue valence. […] Additionally, overall M-atio and M-ias did not correlate with one another (r = 0.37, p = 0.07). These results demonstrate that perceptual and metacognitive biases for noisy stimuli are selectively altered by arousing disgust cues, even in the absence of performance differences in perception or metacognition.”